# Supplementation of *Weizmannia coagulans* BC2000 and Ellagic Acid Inhibits High-Fat-Induced Hypercholesterolemia by Promoting Liver Primary Bile Acid Biosynthesis and Intestinal Cholesterol Excretion in Mice

**DOI:** 10.3390/microorganisms11020264

**Published:** 2023-01-19

**Authors:** Long Jin, Hongyang Dang, Jinyong Wu, Lixia Yuan, Xiangsong Chen, Jianming Yao

**Affiliations:** 1Institute of Plasma Physics, Hefei Institutes of Physical Science, Chinese Academy of Sciences, Hefei 230031, China; 2Science Island Branch, Graduate School of University of Science and Technology of China, Hefei 230026, China; 3Probiotics Institute, Hefei 230031, China; 4College Life Science & Technology, Xinjiang University, Urumqi 830046, China; 5Institute of Nutrition and Health, Qingdao University, Qingdao 266021, China

**Keywords:** ellagic acid, *Weizmannia coagulans*, urolithin A, hypercholesterolemia, metabolomics

## Abstract

The probiotic *Weizmannia coagulans* (*W. coagulans*) BC2000 can increase the abundance of intestinal transforming ellagic acid (EA) bacteria and inhibit metabolic disorders caused by hyperlipidemia by activating liver autophagy. This study aimed to investigate the inhibitory effects of *W. coagulans* BC2000 and EA on hyperlipidemia-induced cholesterol metabolism disorders. C57BL/6J mice (*n* = 10 in each group) were fed a low-fat diet, high-fat diet (HFD), HFD supplemented with EA, HFD supplemented with EA and *W. coagulans* BC77, HFD supplemented with EA, and *W. coagulans* BC2000. EA and *W. coagulans* BC2000 supplementation prevented HFD-induced hypercholesterolemia and promoted fecal cholesterol excretion. Transcriptome analysis showed that primary bile acid biosynthesis in the liver was significantly activated by EA and *W. coagulans* BC2000 treatments. EA and *W. coagulans* BC2000 treatment also significantly increased the intestinal *Eggerthellaceae* abundance and the liver EA metabolites, iso-urolithin A, Urolithin A, and Urolithin B. Therefore, *W. coagulans* BC2000 supplementation promoted the intestinal transformation of EA, which led to the upregulation of liver bile synthesis, thus preventing hypercholesterolemia.

## 1. Introduction

Ellagic acid (EA) is a natural polyphenolic compound that is widely present in fruits and nuts, with biological regulation targeting the gut microbiota [1] and cholesterol metabolism [2,3]. However, the strong polarity of EA complicates its absorption into the digestive tract according to the Lipinski rule of five [4]. Urolithins, gut-microbial-derived metabolites of EA [5], are more easily absorbed than EA [6]. One of the main urolithin isomers present in human plasma and urine is urolithin A, which is the main reason for the physiological impact of EA on the diet [7]. Notably, the intestinal biotransformation of EA into urolithins varies widely among individuals, with some exhibiting high or low conversion rates and others converting nil or very little [8]. Particularly in obese individuals, the structure of the gut microbiota is altered, thereby reducing the efficiency of EA transformation [8,9,10].

Modulation of the gut microbiota can potentially enhance the biological activity of EA. Numerous studies have revealed that probiotics can restructure the gut microbiota [11], thus improving the bioavailability of polyphenols [12]. Two gut species, *Gordonibacter urolithinfaciens* [13] and *Ellagibacter isourolilithinfaciens* [14], have been found to promote the transformation of EA, but are not on the list of species that can be used for food as stipulated in Chinese regulations. This necessitates further investigation of edible probiotics that can transform EA. The exploration of methods for increasing the transformation efficiency of EA through regulation of the gut microbiota is also crucial, and the combined consumption of EA and probiotics may have this effect. *Weizmannia coagulans* (*W. coagulans*), which regulates the gut microbiota, is a probiotic that can be used as food. It is favored for its remarkable probiotic properties (resistance to stomach acid, bile salts, heat, and oxygen, and high shelf-life of the product) [15]. A recent study found that *W. coagulans* can promote the conversion of polyphenols [12]. Our previous study demonstrated that *W. coagulans* BC2000 increased the abundance of intestinal EA-transforming bacteria and inhibited the metabolic disorder caused by hyperlipidemia by activating liver autophagy, a Urolithin A-targeted pathway [16,17]. The potential of a combination of *W. coagulans* BC2000 and EA to increase liver urolithin A levels has not been determined. Urolithin A has been found to improve cholesterol metabolism [4] and reduce cholesterol accumulation in liver cells [18]. This implies that the treatment using a combination of *W. coagulans* BC2000 and EA may regulate cholesterol metabolism through urolithin A, but has yet to be evidenced.

This study aimed to investigate the inhibitory effects of a combination of *W. coagulans* BC2000 and EA on hyperlipidemia-induced cholesterol metabolism disorders. *W. coagulans* BC77, a strain that does not activate liver autophagy, combined with EA, was used as the control. Simultaneously, we used the targeted metabolome to demonstrate the effect of *W. coagulans* BC2000 on the EA metabolites of the intestinal flora in the liver. We also performed plasma non-targeted metabolome and liver transcriptome analyses to clarify the mechanism by which *W. coagulans* BC2000 and EA prevent HFD-induced abnormal cholesterol metabolism.

## 2. Materials and Methods

### 2.1. Reagent

*W. coagulans* BC2000 and *W. coagulans* BC77 were isolated from sauerkraut in Northeast China and stored in the General Microbiology Center of the Microbial Strain Management Committee of China, supplied by Chacha Food Co., Ltd. (Hefei, China). An EA purity of 99.9% was supplied by Yuanye Biotechnology Co., Ltd. (Shanghai, China).

### 2.2. Diets and Animals

The animal experimental protocol for this study was approved by the Animal Ethics Committee of Qingdao University (approval No.: QDU-AEC-2022259). Fifty C57BL/6J male mice (6 weeks old) were purchased from SPF Biotechnology Co., Ltd. (Beijing, China). These mice were reared at the Qingdao University Laboratory Animal Center under 12/12 h light/dark cycles, a constant temperature of 22 ± 1 °C, and a constant humidity level of 50% ± 5%. Adaptive feeding was administered for one week. Fifty mice were randomly divided into five groups (*n* = 10 for each group) to receive a low-fat diet (LFD, 10% energy from fat), high-fat diet (HFD, 45% of energy from lard), HFD + EA, HFD + EA + *W. coagulans* BC77 (EABC77), and HFD + EA + *W. coagulans* BC2000 (EABC2000). The method for preparing the feed is described in Appendix A. Our previous experiments found that the lyophilized powder of *W. coagulans* at 5.0 × 10^7^ CFU/g would reach 4.0 × 10^7^ CFU/g after 9 months (Appendix A). This indicated that the lyophilized probiotic powder was stable. The feed contained lyophilized *W. coagulans* BC2000 or *W. coagulans* BC77 powder, with a probiotic content of 4 × 10^11^ CFU/g. According to the feed intake of 2.3–3.3 g/d/mouse, lyophilized powder was added to the feed at 0.1 g/kg. In other words, the *W. coagulans* BC2000 or *W. coagulans* BC77 intake was 9.2 × 10^7^–1.32 × 10^8^ CFU/d/mouse. Mice were given a 0.3 g EA/kg diet at approximately 30 mg/kg body weight per day based on previous studies [19,20]. The feed formulations used for the mice are described in Appendix A. There were 3–4 mice in each cage for ten weeks. At each feeding, 2–3 days of feed was supplied with 7 days of water. All mice were fed and hydrated ad libitum.

### 2.3. Biochemical Analyses

All mice were anesthetized and euthanized by cervical dislocation after 12 h of fasting following the animal experiments. Blood from the posterior orbital sinus was collected from the mice. Plasma was separated via anticoagulation with ethylenediaminetetraacetic acid using a centrifuge (Sigma-Aldrich, St. Louis, MO, USA) at 1500× *g* and 4 °C for 10 min. Total cholesterol (TC), triglyceride (TG), high-density lipoprotein cholesterol (HDL-C), and low-density lipoprotein cholesterol (LDL-C) levels in the plasma were measured according to the manufacturer’s instructions (Nanjing Jiancheng Bioengineering Institute, Nanjing, China). One day before the mice were euthanized, their feces were collected. Immediately after the mice defecated, fecal samples were placed in Eppendorf tubes and rapidly frozen in a tank of liquid nitrogen. The levels of TC and bile acids (BAs) in the feces were quantified using an enzyme-linked immunosorbent assay kit (Jiangsu Jingmei Bio-technology Co., Ltd., Yancheng, China).

### 2.4. Visceral Organ Indices

Mouse epididymal fat and livers were collected and weighed for visceral organ indexing [21]. The formula used was as follows: visceral organ index = visceral organ weight/body weight.

### 2.5. Analysis of EA and Urolithins by UPLC-ESI-QTOF-MS/MS

According to a previously described method [22], a 100 mg liver sample was weighed, and steel beads were added. They were homogenized with methanol: HCl (99.9:0.1 *v*/*v*) three times at an intensity of 50 Hz on a homogenizer (30 s each time, with a 10 s interval between two homogenizations). The homogenate was centrifuged at 12,000× *g* rpm at 4 °C for 15 min. The supernatant was added to the liquid phase vial by filtering through a 0.45 µm PVDF filter prior to UPLC-ESI-QTOF-MS/MS analysis (where QTOF refers to quadrupole TOF). The analytical methods are described in Appendix A. An internal standard (6,7-dihydroxycoumarin, 0.2 ppm) was added prior to sample preparation to control extraction efficiency, and 10 μL of each sample was collected for quality control (QC).

Compounds were identified based on ionic fragments provided by Pubchem (https://pubchem.ncbi.nlm.nih.gov/ (accessed from 28 August 2022 to 15 September 2022)), HMDB (https://hmdb.ca/ (accessed from 28 August 2022 to 15 September 2022)), and MassBank of North America (https://mona.fiehnlab.ucdavis.edu/ (accessed from 28 August 2022 to 15 September 2022)), and previous studies [23,24]. The data were processed and analyzed using MassHunter Qualitative Analysis software (B.08.00; Agilent Technologies, Waldbronn, Germany). The relative quantification of metabolites identified in MS was achieved by integrating the peak areas of the extracted ion chromatograms, with internal standards correcting for peak areas. 

### 2.6. 16S rRNA Sequencing of Microbiota in the Cecum Content

Total DNA was extracted from the cecum content using the Power Soil DNA Isolation Kit (MO BIO Laboratories) [25]. The microbiota was analyzed using 16S rRNA amplicon sequencing PCR primers (forward primer: 5′- CCT ACG GRR BGC ASC AGK VRV GAA T -3′; reverse primer: 5′- GGA CTA CNV GGG TWT CTA ATC C-3′) to amplify the V3 and V4 hypervariable regions, followed by the addition of an index-link at the end of the 16S rDNA PCR product. The PCR-amplified library was subjected to PE250/PE300 paired-end sequencing using the Illumina MiSeq system [26]. EasyAmplicon v1.14 was used to perform downstream amplicon bioinformatic analysis. Species-level annotation was accomplished using the Bayesian-lowest-common-ancestor (BLCA) algorithm [27], based on the NCBI 16S Microbial database (https://www.ncbi.nlm.nih.gov/ (accessed from 1 December 2021 to 8 December 2021) ).

MicrobiomeAnalyst (http://www.microbiomeanalyst.ca/ (accessed from 9 December 2021 to 10 January 2022) ) and ImageGP (http://www.ehbio.com/ImageGP/ (accessed from 9 December 2021 to 10 January 2022) ) were used to plot the analysis results. For gut microbiota taxon analysis, alpha diversity measures included Chao1 and Shannon indices. Differences among groups at the OTU level were analyzed using principal coordinate analysis (PCoA) of unweighted UniFrac distances and permutational multivariate analysis of variance (PERMANOVA). Species-level differential bacteria between the groups were identified using linear discriminant analysis (LDA) effect size (LEfSe) (FDR < 0.05, LDA = 4).

### 2.7. Determination of Cecum SCFAs 

According to a previous experimental method [28], aqueous solutions of acetic acid, propionic acid, and butyric acid were extracted from cecum content samples and tested by GC-MS (Agilent 8890 GC System, Waldbronn, Germany). The analytes were quantified using standard curves obtained by dilution, extraction, and derivatization of the standards.

### 2.8. Plasma Metabolomics by UPLC-QTOF-MS

For each sample, 50 μL of plasma was added to 750 μL of precooled extraction buffer (methanol: ultrapure water = 1:1). After vortexing, samples were incubated on ice for 15 min. The mixture was centrifuged at 12,000× *g* rpm at 4 °C for 15 min. The supernatant was collected in a liquid-phase chromatography vial. In addition, 10 μL of each prepared sample was pooled as the QC sample. Untargeted Metabolomics was performed using an Agilent 1290 Infinity UPLC system connected to a 6530 B Q-TOF mass spectrometer. Chromatographic and mass spectrometry conditions are described in the Appendix A. To correct the data deviations caused by the instrument state during testing, the QC sample was analyzed once every eight samples were tested. Sample testing and instrument operation were monitored using Data Acquisition Software.

MS-DIAL 4.38 was used for the raw metabolite data processing. The metabolites were identified and analyzed based on the HMDB (https://hmdb.ca/ (accessed from 12 February 2022 to 1 April 2022) ), METLIN (https://metlin.scripps.edu/landing_page.php?pgcontent=mainPage (accessed from 12 February 2022 to 1 April 2022) ), and PubChem (https://pubchem.ncbi.nlm.nih.gov/ (accessed from 12 February 2022 to 1 April 2022) ) databases. Integrated mass-spectrometry-based untargeted metabolomics data mining (IP4M) software [29] was used for peak table pre-processing and orthogonal partial least squares discriminant analysis (OPLS-DA). Based on the variable importance in the projection (VIP) threshold of 1 from the OPLS-DA model, differential metabolites in the metabolic profiles between HFD and LFD, EA and HFD, EABC77 and HFD, and EABC2000 and HFD were obtained.

### 2.9. RNA Sequencing and Gene Set Enrichment Analysis

Total RNA was extracted from the liver for transcriptomic analysis. The RNA expression profiles were normalized using the R package “limma”. Differentially expressed genes (DEGs) were identified using Bayesian-adjusted t-statistics of limma’s linear model (log (fold change) (log2FC) >1.5, *p <* 0.05). Gene enrichment analysis of the KEGG pathway was performed on DEGs using the R package “Clusterprofiler v4.0”. KEGG pathways were visualized using the Pathview software. To acquire protein–protein interaction (PPI) networks, DEGs were mapped to the search tool for the retrieval of interacting genes (STRING) (version 11.5) online database [30]. The highest confidence level in the argument for interactions was set at >0.4.

### 2.10. Statistical Analysis

The mean differences between the groups were analyzed using one-way analysis of variance (ANOVA) and Tukey’s test, or the Kruskal–Wallis test. Data were statistically analyzed using SPSS 23.0 and expressed as mean ± SEM. *p* < 0.05 or *p* < 0.01 were regarded as being statistically significant. Heat maps were visualized using HEML 1.0. All other figures were created using GraphPad Prism 8.0.

## 3. Results

### 3.1. Joint Consumption of W. coagulans BC2000 and EA for the Prevention of Abnormal Lipid Metabolism in Mice Fed an HFD 

From week 5 to week 10, the body weight of the mice in the HFD group was significantly higher than that of the mice in the LFD group (*p* < 0.05, Figure 1B). Following EA supplementation, body weight decreased significantly (*p* < 0.05, Figure 1B). EABC77 and EABC2000 interventions did not induce obesity. Mice that received an HFD diet consumed less food than those that received an LFD diet (*p* < 0.05, Figure 1C). As shown in Figure 1D,E, the adiposity and liver indices were significantly higher in the HFD group than in the LFD group, and EA, EABC77, and EABC2000 treatments showed a trend toward remission, but this was not statistically significant. Figure 1F–J show a significant increase in plasma levels of TC, TG, LDL-C, and LDL-C/HDL-C in the HFD group (*p* < 0.05). However, supplementation with EA and EA + *W. coagulans* effectively alleviated the HFD-induced elevation of TG and TC levels caused by HFD, especially EABC2000. Mice fed with EABC2000 showed 38% and 17% reductions in plasma TC levels compared to mice fed HFD and EA diets, respectively. In addition, the plasma TC levels in the EABC77 group were not significantly different from those in the EA group. These findings showed that combined interventions were superior to the sole administration of EA. EABC2000 and EABC77 interventions prevented HFD-induced increases in TC, TG and LDL-C in mice, and *W. coagulans* BC2000 was more effective in preventing the elevation of TC caused by an HFD than *W. coagulans* BC77. As a result, EABC2000 can prevent hypercholesterolemia caused by an HFD.

### 3.2. W. coagulans Promotes the Transformation of EA into Urolithin in the Liver of Mice

The UPLC-ESI-QTOF-MS/MS chromatograms of EA metabolites in mouse liver and MS/MS mass spectra of the identified compounds are detailed in the Appendix A (Appendix A). Urolithins, the metabolites of EA, are the main cause of the physiological impact of EA [11]. Seven urolithins were identified in the mouse liver, including urolithin A, iso-urolithin A, urolithin B, and other urolithin compounds. EA was not identified in the liver, indicating its limited absorption. As shown in Table 1, the levels of urolithin A in the livers of mice in the EABC2000 group were 5.20-fold those in the EA group and 2.58-fold those in the EABC77 group. Urolithin A was significantly enriched in the EABC2000 group (*p* < 0.05). Compared with *W. coagulans* BC77, *W. coagulans* BC2000 can better facilitate the metabolism of EA to urolithin via the gut microbiota, thereby enhancing the physiological effects of EA.

### 3.3. Joint Consumption of W. coagulans BC2000 and EA to Reshape the Cecum Microbiota in Mice

To investigate changes in the microbiota in the cecum contents, α-diversity was assessed using the Cho1 and Shannon indices. The Cho1 index was significantly lower in the EABC2000 and EA groups than in the HFD group (*p* < 0.05, Figure 2A). HFD resulted in a notable decrease in the Shannon index, which was alleviated by EABC2000 intervention without statistical significance (Figure 2B). The beta diversity of the cecum microbiota was significantly altered by the five treatments, with the EABC2000 treatment group deviating significantly from the HFD and EA groups (*p* < 0.05, Figure 2C). The *Firmicutes* to *Bacteroidetes* abundance ratio increased in the HFD groups, which was alleviated by EABC2000 treatment (*p* < 0.05, Figure 2D). At the genus level, the relative abundance of *Lactobacillus* was remarkably elevated in the EABC2000 intervention group compared with that in the LFD group (*p* < 0.05, Figure 2E). At the species level, the relative abundance of *Muribaculum intestinale* [31], which participates in carbohydrate metabolism, was significantly increased in the EABC2000 intervention group (*p* < 0.05, Figure 2F). LEfSe analysis showed significantly lower *Desulfovibrio vulgeris* of opportunistic pathogenic bacteria in the EABC2000 intervention group (FDR < 0.05, LDA score ≥ 4, Figure 2G). *W. coagulans* BC77 and *W. coagulans* BC2000 colonized the intestine of mice. *W. coagulans* BC2000 had a more optimized colonization capacity than *W. coagulans* BC77 (*p* < 0.05, Figure 2H). *Eggerthellaceae* was significantly enriched in the EABC2000 group (*p* < 0.05, Figure 2I). *Eggerthellaceae* is associated with the degradation of polyphenols [32] and is a potential family for EA transformation. These findings showed that EABC2000 intervention can reshape the cecum microbiota and promote the enrichment of the *Eggerthellaceae* family.

### 3.4. Joint Consumption of W. coagulans BC2000 and EA Does Not Regulate the Metabolism of SCFAs in Mice

Compared with the LFD group, acetate in the cecum contents of mice in the HFD group was reduced (Figure 3A). Propionate levels in the HFD group were significantly lower than those in the LFD group (*p* < 0.05) and were upregulated after EABC2000 treatment, but the difference was not statistically significant (Figure 3B). The EABC2000 intervention group also showed increased butyrate levels, although this was not statistically significant (Figure 3C). However, it significantly improved butyric acid metabolism in the microbiota. These findings suggest that intervention with EABC2000 does not significantly regulate the metabolism of SCFAs.

### 3.5. Joint Consumption of W. coagulans BC2000 and EA Alters Plasma Metabolites in Mice

Untargeted metabolomic analysis of plasma showed that the metabolite profile of the HFD group deviated from that of the LFD group, according to OPLS-DA (Figure 4A). EA, EABC77, and EABC2000 treatments caused significant changes in plasma metabolic profiles (EA: Figure 4D; EABC77: Figure 4G; EABC2000: Figure 4J), and the model was remarkable (EA: R^2^Y = 0.987, Q^2^Y = 0.547, Figure 4E; EABC77: R^2^Y = 0.974, Q^2^Y = 0.685, Figure 4H; EABC2000: R^2^Y = 0.981, Q^2^Y = 0.667, Figure 4K). OPLS-DA V-plots for HFD versus LFD (Figure 4C), EA versus HFD (Figure 4F), EABC77 versus HFD (Figure 4I), and EABC2000 versus HFD (Figure 4L) were constructed to select the significant metabolites that were differentially expressed between the two groups, and variables with a VIP exceeding 1 were selected for subsequent analysis. Plasma butyric acid, taurocholic acid, tauroursodeoxycholic acid, glycocholic acid, leucine, l-valine, and kynurenine were significantly downregulated in the HFD group. Plasma 7z,10z-hexadecadienoic acid and retinoic acid were significantly upregulated, and plasma leucine and norvaline were significantly downregulated after EABC2000 treatment. However, after EABC77 treatment, leucine and norvaline levels were significantly downregulated (*p* < 0.05, Table 2). Untargeted metabolomic analysis showed that EABC2000 could regulate the composition of plasma metabolites associated with obesity and inflammation.

### 3.6. Joint Consumption of W. coagulans BC2000 and EA Promotes Hepatic BA Biosynthesis and Cholesterol Excretion in Mice

We further investigated how the combined consumption of EA and *W. coagulans* BC2000 lowered plasma cholesterol levels. We examined the fecal levels of TC and total BAs. BAs in feces are the main products of cholesterol metabolism. Increased fecal BAs excretion can reduce plasma cholesterol levels [33]. As shown in Figure 5A,B, the HFD resulted in significantly lower fecal excretion of TC and total BAs (*p* < 0.05). The levels of TC and total BAs in feces significantly increased after EABC2000 treatment (*p* < 0.05). However, EABC77 treatment promoted only the excretion of total BAs (*p* < 0.05).

Subsequently, mouse livers were transcriptome-sequenced to investigate the role played by *W. coagulans* BC2000 in preventing elevated TC in mice on an HFD. EABC2000 treatment significantly increased the expression of *Cyp7a1* (*p* < 0.05, Figure 5C) and increased the expression of *Abca1* (Figure 5E) compared to the HFD group. In addition, EABC2000 intervention reduced the expression of *S1c10a2*, but the difference was not statistically significant (Figure 5D). KEGG enrichment analysis of DEGs from EABC2000 versus from EA revealed that metabolic pathways associated with BAs biosynthesis and metabolism were activated. These included primary bile acid biosynthesis, steroid hormone biosynthesis, and taurine and hypotaurine metabolism (Figure 5F). PPI networks verified that the activation of primary BAs biosynthesis was closely associated with the activation of *Cyp46a1*, *Cyp7a1*, *Cyp7b1*, and *Akr1d1* (Figure 5H). In addition, as shown in Figure 5G, compared to EA treatment, cholesterol 24-hydroxylase, cholesterol 7alpha-monooxygenase, and 3-oxo-5-beta-steroid 4-dehydrogenase were significantly activated after EABC2000 treatment. 26-hydroxycholesterol 7alpha-hydroxylase was significantly inhibited after EABC2000 treatment. These findings showed that the reduction in plasma cholesterol in high-fat mice was closely related to EABC2000 intervention in regulating the metabolism of BAs and promoting cholesterol excretion.

### 3.7. Gut Microbiota Is Correlated with Indicators Related to Lipid Metabolism

As shown in Figure 6, *Desulfovibrio* was significantly and positively correlated with TG, TC, LDL-C, and LDL-C/HDL-C levels (*p* < 0.01). *Lactobacillus* was significantly and negatively correlated with tauroursodeoxycholic and glycocholic acid levels (*p* < 0.05). *Muribaculum intestinale* was significantly and negatively associated with TG, TC, LDL-C, HDL-C, and LDL-C/HDL-C levels (*p* < 0.05, *p* < 0.01). *Firmicutes/Bacteroides* were significantly and positively correlated with TC, TG, LDL-C, HDL-C, and LDL-C/HDL-C (*p* < 0.001), and significantly and negatively correlated with total TC (fecal) and BAs (*p* < 0.05, *p* < 0.001). These results showed that the gut microbiota is significantly associated with indicators related to lipid metabolism and that modulating the gut microbiota may prevent hypercholesterolemia caused by an HFD.

## 4. Discussion

In the present study, EABC2000 intervention prevented HFD-induced hypercholesterolemia by modulating the gut microbiota, activating primary BAs biosynthesis in the liver, and promoting fecal cholesterol excretion. This may be attributed to the increase in gut-microbiota-derived EA metabolites in the liver: urolithin A, urolithin B, and iso urolithin A. A large body of cellular, animal, and human data supports the role of urolithins in the regulation of glucolipid metabolism [18,34,35] and the gut microbiota [36].

Previous studies have shown that disorders of blood lipid metabolism can be ameliorated by EA [37] or EA-rich food [38,39]. EA can reduce plasma levels of TG, TC, and LDL-C [40], [41]. Our previous studies have demonstrated that the combination of *W. coagulans* and EA inhibits metabolic disorders caused by a high-fat diet by activating hepatic autophagy, a urea-A-targeted pathway [16]. However, whether *W. coagulans* BC2000 or EA can increase hepatic urolithin A levels is unclear. In the present study, our results showed that a mixture of *W. coagulans* BC2000 and EA increased the levels of metabolites of EA in the liver of mice, particularly urolithin A, urolithin B, and isourolithin A. The combined consumption of *W. coagulans* BC2000 and EA showed more optimized lipid-lowering effects than EA alone. Urolithin, a metabolite of ellagic acid in the intestinal tract, has more optimized bioavailability. Urolithin A significantly reduced triglyceride accumulation in hepatocytes and adipocytes [18]. Urolithin B reduces lipid plaque deposition in double-knockout ApoE mice [42]. In addition, a previous study showed that the hypolipidemic effects of urolithin A and urolithin B occur through increased clearance of hepatic and serum cholesterol secreted in the bile and via increased excretion of hepatic lipids in the feces [43], which is consistent with our findings. We found that EA, EABC77, and EABC2000 treatments similarly reduced plasma TC, TG, and LDL-C levels, and EABC2000 treatment significantly increased fecal cholesterol and bile acid levels. Therefore, we hypothesized that *W. coagulans* BC2000 facilitated the conversion of EA to urolithin.

In addition, we investigated whether the *W. coagulans* BC2000-promoted conversion of EA to urolithin was associated with changes in the gut microbiota. The active substance urolithin produced by EA can be metabolized by the colonic microbiota [44]. Studies have demonstrated that the production of urolithins may be affected by changes in gut microbiota [8]. Excessive HFD results in dysbiosis of the gut microbiota, which adversely affects action [45]. Recently, it has been proposed to stratify individuals according to their urinary urolithin excretion status [6]. Three metabolic phenotypes associated with the gut microbiota were defined. Metabotype A includes producers of only urolithin A; metabotype B includes subjects who produce urolithin A, urolithin B, and iso-urolithin A; and metabotype 0 includes individuals that cannot produce final urolithin. EA and urolithins showed complex biological actions including the modulation of many important signaling pathways involved in inflammation and aging. Thus, metabolic phenotype 0 could most benefit from the diet addition of EA and probiotic *Gordonibacter urolithinfaciens* [46]. Another study confirmed that gut bacterium *Gordonibacter urolithinfaciens* metabolized EA into urolithins M5, M6, and C through dehydroxylation with the involvement of nicotinamide adenine dinucleotide phosphate (NADPH) and flavin adenine (FAD) [47]. *Gordonibacter urolithinfaciens* converted EA into urolithins M5, M6, and C, and was positively correlated with the metabotype A. The recently discovered *Ellagibacter isourolithinifaciens* was also able to produce iso-urolithin A and positively correlated with metabotype B [48]. *Gordonibacter urolithinfaciens* and *Ellagibacter isourolithinifaciens,* belonging to the *Eggerthellaceae* family, are two human gut bacterial species that convert EA into urolithins. In this study, the abundance of *Eggerthellaceae* was higher in the EABC2000 group than in the EABC77 and EA groups. *Gordonibacter urolithinfaciens* and *Ellagibacter isourolithinifacien* could promote the conversion of EA to urolithin, which enters the systemic circulation through the bloodstream and, thus, plays a physiological role. Therefore, urolithin A and urolithin B were significantly enriched in the EABC2000 group. These results suggest that the facilitation of EA conversion to urolithin by *W. coagulans* BC2000 is closely related to alterations in the gut microbiota, particularly the enrichment of the *Eggerthellaceae* family.

The gut microbiota is widely identified as a potential target for the treatment of hyperlipidaemia [49], where dietary interventions are an important factor in modulating its composition [50]. In the present study, the Chao1 and Shannon indices of the mouse microbiota were altered after EABC2000 treatment, indicating that EABC2000 treatment had a positive effect on the abundance and diversity of the gut microbiota. Increased *Firmicutes*/*Bacteroidetes* (F/B) ratios in the gut microbiota of HFD mice are thought to be characteristic of obesity and metabolic syndrome in humans and animal models [51]. Previous studies have shown that EA can reduce the F/B ratio and, thus, improve lipid metabolism in hyperlipidaemia-induced mice [52]. In this study, the EABC2000 intervention was more effective in reducing the F/B ratio than EA intervention alone. These results suggest that EABC2000 plays a beneficial role by modulating the ratio and diversity of gut microbiota associated with the development of hyperlipidemia.

Certain gut microbiota may influence the development and incidence of hyperlipidemia. Notably, EABC2000 treatment reshaped the gut microbiota. *Faecalibaculum* is a known SCFA-producing bacterium that has been reported to improve the intestinal stress response [53]. It was the main genus in the EA, EABC77, and EABC2000 treatment groups. *Lactobacillus* improves host physiology and lipid metabolism. It also prevents chronic inflammation by maintaining the intestinal barrier [54]. In addition, certain studies have shown that the abundance of the *Lactobacillus* genus is significantly increased in the gut of urolithin producers [55]. In this study, the *Lactobacillus* genus was significantly enriched in the EABC2000 group. A typical Western diet (high fat and sugar) results in a decreased abundance of the beneficial gut bacterium *Muribaculum intestinale* in mice [31]. This is consistent with the results of the present study, in which *Muribaculum intestinale* was significantly enriched after EABC2000 treatment. *Muribaculum intestinale* has been proven to be involved in the metabolism of carbohydrates [31] and produces propionate [56]. Propionate has physiological activities that lower cholesterol, reduce fat storage, and have anticancer and anti-inflammation [57]. Importantly, we observed a significant negative correlation between *Muribaculum intestinale* and TC, TG, LDL-D, and HDL-C levels. Therefore, these bacteria could be used as biomarkers for the treatment of mice and as intestinal indicators to monitor HFD-induced hypercholesterolemia. However, further studies are required to determine the role of bacteria in the regulation of EABC2000. In addition, the growth of the opportunistic pathogen *Desulfovibrio* was inhibited after EABC2000 treatment compared with EABC77. Moreover, *Desulfovibrio* positively correlated with TC, TG, LDL-D, and HDL-C. It was reported that *Desulfovibrio* could produce endotoxins such as lipopolysaccharide (LPS), which are associated with the development of intestinal diseases [58]. In summary, EABC2000 modified HFD-induced hypercholesterolemia in mice by reshaping intestinal microbiota.

In addition, there are mutually beneficial interactions between phenolic compounds and probiotics, and probiotics and gut flora improve the metabolism and bioavailability of phenolic compounds, while phenolic compounds can modulate the gut flora by increasing the adhesion and survival of probiotics [59]. *W. coagulans* was significantly enriched in the EABC2000 group. This may be due to the easier colonization of the gut by *W. coagulans* BC2000 than by *W. coagulans* BC77 and the beneficial effect of ellagic acid on *W. coagulans* colonization; however, verification is required.

In the liver, BAs are the main pathway for cholesterol catabolism, accounting for 50% of the daily cholesterol [60]. The BAs biosynthetic pathway involves two main pathways: the classical pathway and alternative pathway [61]. *Cyp7a1* is a key enzyme that catalyzes the first rate-limiting step in the classical BAs biosynthesis pathway [62]. It has been reported that deficiency of *Cyp7a1* leads to hypercholesterolemia [63], while plasma cholesterol levels are reduced in mice overexpressing *Cyp7a1* [64]. An HFD leads to higher cholesterol levels in the body. EA can promote cholesterol clearance through two pathways: enhancement of BAs in feces and activation of the *Lxr*/*Ppar*-*Abac1* signaling pathway [3]. Urolithin A regulates *Cyp7a1* expression to inhibit lipid accumulation [65]. These findings are consistent with the results of the present study. In this study, we found that EABC2000 supplementation significantly regulated the expression of *Cyp7a1* and BAs biosynthesis-related enzymes in the mouse liver and increased fecal cholesterol and BAs levels. As expected, EABC2000 treatment caused an increase in BAs excretion, which could subsequently lower their concentration, forcing the upregulation of the hepatic de novo synthesis of BAs from endogenous cholesterol by activating *Cyp7a1* [64]. Furthermore, our liver transcriptomic and plasma metabolomic results showed that EABC2000 supplementation significantly promoted primary BAs biosynthesis, suppressed the expression level of the BA reabsorption transporter gene *S1c10a2* without significant differences, and did not alter the HFD-induced reduction in taurocholic, tauroursodeoxycholic, and glycocholic acids. This suggests that EABC2000 supplementation did not exert a cholesterol-lowering effect by modulating the promotion of BA reabsorption, but rather by increasing fecal BA excretion to lower plasma cholesterol. In other words, the increase in fecal BA excretion after EABC2000 treatment, followed by a decrease in body BAs concentration, forced the liver to promote the biosynthesis of primary BAs by activating the expression of *Cyp7a1*, the rate-limiting enzyme for BAs biosynthesis, resulting in decreased plasma cholesterol levels (Figure 7). In addition, we found that fecal cholesterol levels were elevated in mice after EABC2000 treatment. This may be due to increased cholesterol excretion from the host after EABC2000 treatment.

## 5. Conclusions

In summary, *W. coagulans* BC2000 promotes the conversion of EA to urolithin and helps EA exert its physiological effects upon joint consumption. A combination of EA and *W. coagulans* BC2000 prevents HFD-induced hypercholesterolemia by promoting bile acid biosynthesis, cholesterol excretion, and remodeling of the gut microbiota. This study offers new ideas for improving the physiological effects of EA and provides more evidence that *W. coagulans* BC2000 can be used as a probiotic in foods high in EA. However, the extent to which EA and EA metabolites are absorbed in the body has not been quantified, which is one of the limitations of the study. This presents scope for a further, future study.

## Figures and Tables

**Figure 1 microorganisms-11-00264-f001:**
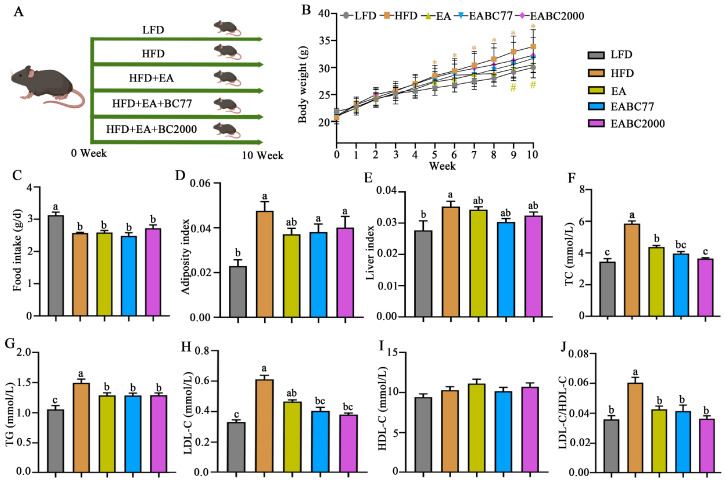
Effect of jointly consuming *Weizmannia coagulans* BC2000 and ellagic acid on lipid metabolism of mice. (**A**) Design of experiment; (**B**) body weight; (**C**) food intake; (**D**) adiposity index; (**E**) liver index; (**F**) TC; (**G**) TG; (**H**) LDL-C; (**I**) HDL-C; (**J**) LDL-C/HDL-C. * indicates a significant difference compared to the LFD group (*p* < 0.05); # indicates a significant difference compared to the HFD group (*p* < 0.05). *n* = 10 in each group. Different letters indicate significant differences between groups by Tukey’s test (*p* < 0.05). LFD: low-fat-diet group; HFD: high-fat-diet group; EA: ellagic acid intervention group; EABC77: EA + *Weizmannia coagulans* BC77 intervention group; EABC2000: EA + *Weizmannia coagulans* BC2000 intervention group. HDL-C: high-density lipoprotein cholesterol; LDL-C: low-density lipoprotein cholesterol; TC: total cholesterol; TG: triglyceride.

**Figure 2 microorganisms-11-00264-f002:**
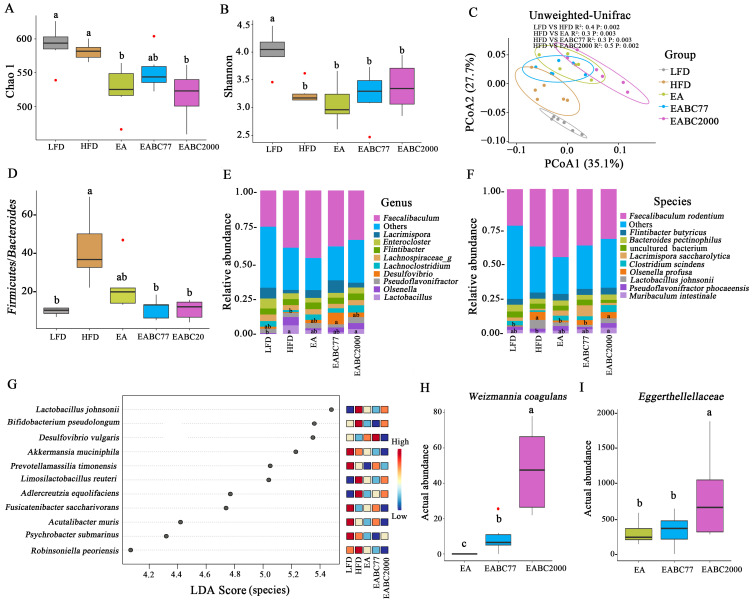
Effect of jointly consuming *Weizmannia coagulans* BC2000 and ellagic acid on cecum microbiota of mice. (**A**) Chao1 index; (**B**) Shannon index; (**C**) Principal coordinate analysis (PCoA) score plots; (**D**) *Firmicutes/Bacteroides* ratio; (**E**) relative abundance of the genus; (**F**) relative abundance of the species; (**G**) linear discriminant analysis (LDA) combined with effect size (LEfSe) measurements at the species level (FDR < 0.05 and LDA score ≥ 4); (**H**) actual abundance of *Weizmannia coagulans* by Wilcoxon test; (**I**) actual abundance of *Eggerthellaceae* by Wilcoxon test. Different letters indicate significant differences between groups by Tukey’s test or Kruskal–Wallis test (*p* < 0.05); *n* = 6 in each group; LFD: low-fat-diet group; HFD: high-fat-diet group; EA: ellagic acid intervention group; EABC77: EA + *Weizmannia coagulans* BC77 intervention group; EABC2000: EA + *Weizmannia coagulans* BC2000 intervention group.

**Figure 3 microorganisms-11-00264-f003:**
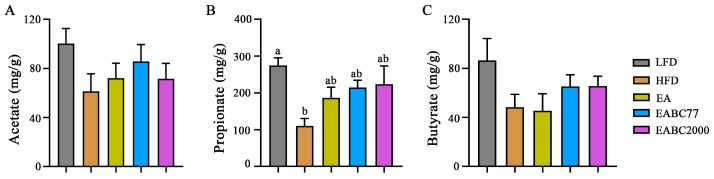
Effect of jointly consuming *Weizmannia coagulans* BC2000 and ellagic acid on short-chain fatty acid of mice. (**A**) Acetate; (**B**) propionate; (**C**) butyrate. Different lowercase letters above the bars represent significant inter-group differences by Tukey’s test or the Kruskal–Wallis test (*p* < 0.05). *n* = 6 in each group. LFD: low-fat-diet group; HFD: high-fat-diet group; EA: EA intervention group; EABC77: EA + *Weizmannia coagulans* BC77 intervention group; EABC2000: EA + *Weizmannia coagulans* BC2000 intervention group.

**Figure 4 microorganisms-11-00264-f004:**
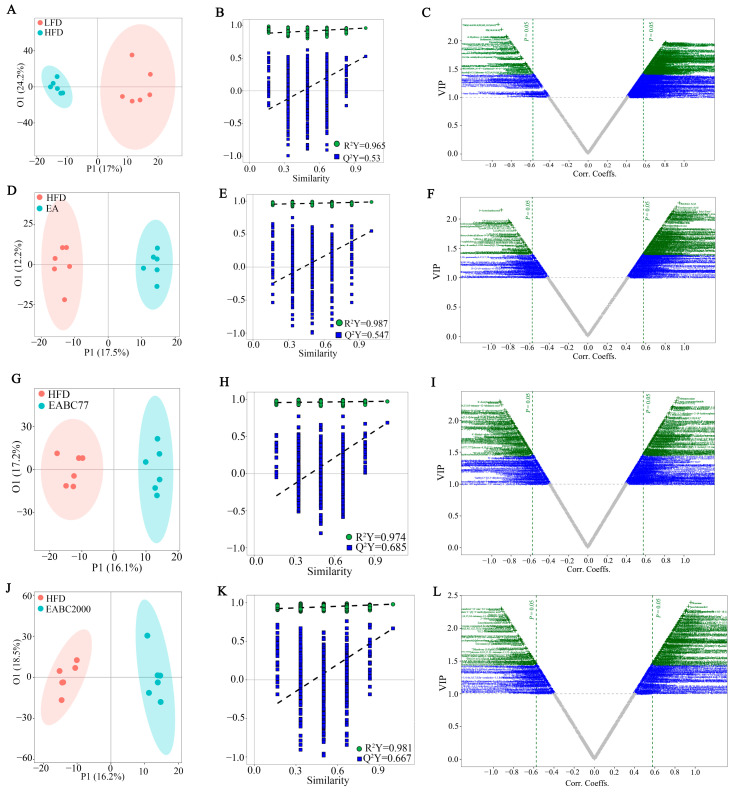
Untargeted metabolomic analysis of mouse plasma metabolites by UPLC-QTOF MS/MS. OPLS-DA score plot (**A**: R2Y = 0.965, Q2Y = 0.53; **D**: R2Y = 0.987, Q2Y = 0.547; **G**: R2Y = 0.974, Q2Y = 0.685; **J**: R2Y = 0.981, Q2Y = 0.667), the corresponding permutation plots (**B**,**E**,**H**,**K**), and v-plot (**C**,**F**,**I**,**L**). LFD: low-fat-diet group; HFD: high-fat-diet group; EA: ellagic acid intervention group; EABC77: EA + *Weizmannia coagulans* BC77 intervention group; EABC2000: EA + *Weizmannia coagulans* BC2000 intervention group.

**Figure 5 microorganisms-11-00264-f005:**
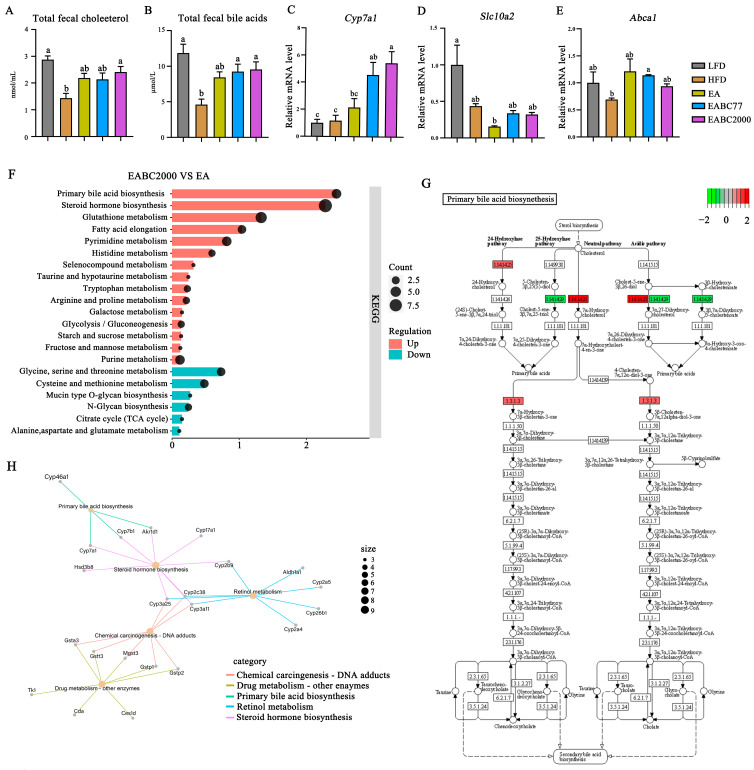
Effect of jointly consuming *Weizmannia coagulans* BC2000 and ellagic acid on cholesterol metabolism. (**A**) Total fecal cholesterol; (**B**) total fecal bile acids; (**C**) relative expression of gene *Cyp7a1*; (**D**) relative expression of gene *S1c10a2*; (**E**) relative expression of gene *Abca1*; (**F**) differential gene KEGG enrichment analysis; (**H**) protein–protein interaction networks map based on differentially expressed genes; (**G**) KEGG pathway map of primary bile acid biosynthesis. Different lowercase letters above the bars represent significant inter-group differences by Tukey’s test (*p* < 0.05). *n* = 4 in each group (total fecal cholesterol and total fecal bile acids, *n* = 5 in each group). LFD: low-fat-diet group; HFD: high-fat-diet group; EA: EA intervention group; EABC77: EA + *Weizmannia coagulans* BC77 intervention group; EABC2000: EA + *Weizmannia coagulans* BC2000 intervention group.

**Figure 6 microorganisms-11-00264-f006:**
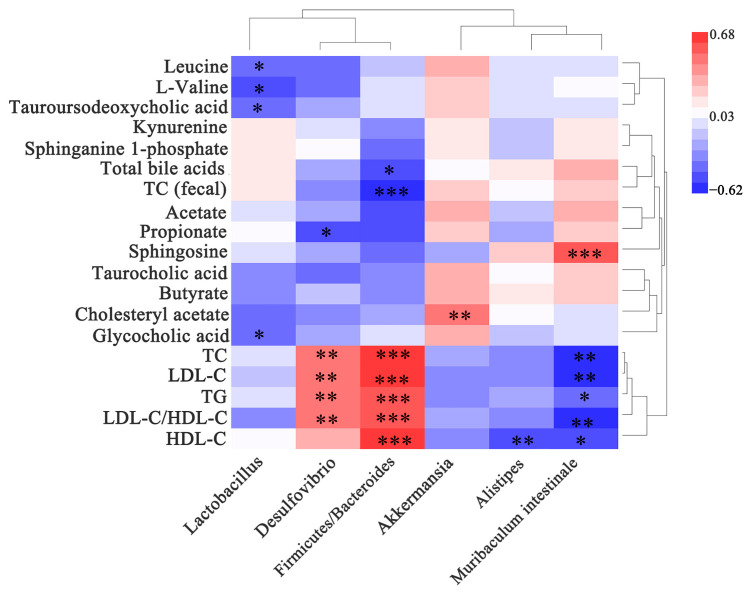
Heat map of the correlation between biochemical indicators and metabolites with gut microbiota. The x-axis of the heat map represents the gut microbiota. The y-axis represents the biochemical indicators and metabolites. The R and *p* values were calculated, and the R values are shown in different colors. The color card on the right shows the range of colors for the different R values. The symbol “*” indicates a significant correlation at the 0.05 level; the symbol “**” indicates a significant correlation at the 0.01 level; the symbol “***” indicates a significant correlation at the 0.001 level. The top clusters represent the clustering of gut microbiota, and the right-side clusters represent the clustering of biochemical indicators and metabolites.

**Figure 7 microorganisms-11-00264-f007:**
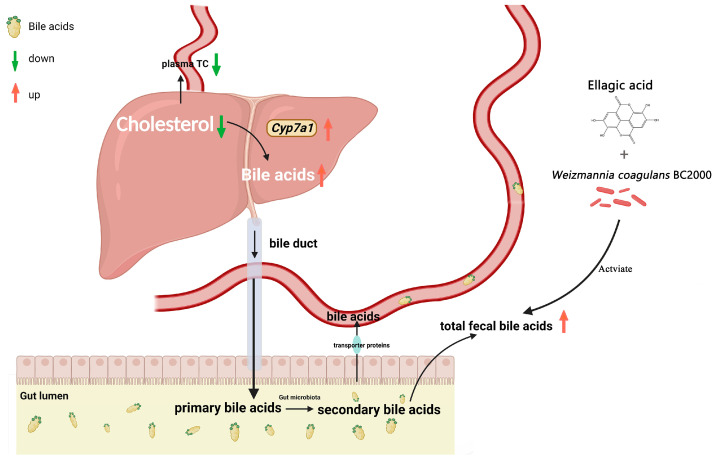
Proposed mechanism of the cholesterol-lowering effect of EABC2000 supplementation. LFD: low-fat-diet group; HFD: high-fat-diet group; EA: EA intervention group; EABC77: EA + *Weizmannia coagulans* BC77 intervention group; EABC2000: EA + *Weizmannia coagulans* BC2000 intervention group.

**Table 1 microorganisms-11-00264-t001:** EA metabolites identified by UPLC-ESI-QTOF-MS/MS analysis.

Metabolites	RT (min)	M/Z [M-H]-	MS/MS Fragments	EABC2000 VS EA		EABC77 VS EA		EABC2000 VS EABC77	
				Log2FC ^a^	*p* ^b^	Log2FC ^a^	*p* ^b^	Log2FC ^a^	*p* ^b^
Urolithin-B sulphate	7.10	290	211	2.12 ↑	0.84	1.41 ↑	0.12	1.08 ↑	0.69
Iso-urolithin A glucuronide	12.81	403	227	2.11 ↑	0.22	1.26 ↑	0.09	1.27 ↑	0.69
Urolithin-C isomer	13.03	243	163, 183, 199	3.07 ↑	0.20	1.16 ↑	0.06	2.16 ↑	0.55
Urolithin-A diglucuronide	14.43	579	175, 227	2.24 ↑	0.84	1.35 ↑	0.16	0.70 ↑	1.00
Iso-urolithin A	17.52	227	159, 171	2.71 ↑	0.84	0.55 ↑	0.22	1.72 ↑	1.00
Urolithin A	18.05	227	160, 183	5.20 ↑	0.01 *	3.81 ↑	0.01*	2.58 ↑	0.03 *
Urolithin B	20.29	211	139, 167	3.83 ↑	0.02 *	0.85 ↑	0.02*	1.39 ↑	0.15

Note: ^a^ Log2 fold change (Log2FC) was obtained by Wilcoxon test; ^b^
*p*-value was obtained by Wilcoxon test; * indicates that *p* < 0.05 or *p* < 0.01 was considered statistically significant by Wilcoxon test. ↑ means up-regulation; ↓ means down-regulation. *n* = 5 in each group. EA: ellagic acid intervention group; EABC77: EA + *Weizmannia coagulans* BC77 intervention group; EABC2000: EA + *Weizmannia coagulans* BC2000 intervention group.

**Table 2 microorganisms-11-00264-t002:** Differential metabolites in the mouse plasma.

Metabolites	HFD vs. LFD		EA vs. HFD		EABC77 vs. HFD		EABC2000 vs. HFD	
	Log2FC ^a^	VIP ^b^	Log2FC ^a^	VIP ^b^	Log2FC ^a^	VIP ^b^	Log2FC ^a^	VIP ^b^
Adrenic acid	0.66 ↑	1.86	0.64 ↑	1.99	-	-	0.50 ↑	1.96
Docosapentaenoic acid	3.03 ↑	1.79	-	-	-	-	-	-
Butyric acid	−0.66 ↓	1.52	-	-	-	-	-	-
Eicosadienoic acid	-	-	0.99 ↑	2.16	0.57 ↑	2.03	0.72 ↑	2.11
Myristic acid	-	-	0.64 ↑	2.08	0.45 ↑	1.77	0.54 ↑	1.99
Alpha-Linolenic acid	-	-	0.93 ↑	1.93	-	-	0.65 ↑	1.72
Arachidonic acid	-	-	0.52 ↑	1.55	-	-	0.46 ↑	1.69
Docosahexaenoic acid	-	-	0.52 ↑	1.43	-	-	-	-
Eicosapentaenoic acid	-	-	0.54 ↑	1.39	-	-	-	-
7Z,10Z-Hexadecadienoic acid	-	-	1.24 ↑	1.99	-	-	0.83 ↑	1.83
Palmitic acid	-	-	-	-	−0.42 ↓	1.70	−0.66 ↓	2.21
Stearoylcarnitine	-	-	-	-	-	-	−0.48 ↓	1.82
Tetradecanoylcarnitine	-	-	−0.52 ↓	1.92	-	-	-	-
3, 5-Tetradecadiencarnitine	-	-	-	-	-	-	−0.38 ↓	1.69
Linoleoyl ethanolamide	-	-	0.80 ↑	1.53	-	-	-	-
L-Palmitoylcarnitine	-	-	-	-	-	-	−0.28 ↓	1.56
2-Methylacetoacetyl-CoA	-	-	-	-	-	-	1.03 ↑	1.50
PS (18:0/18:0)	1.11 ↑	2.19	-	-	0.35 ↑	1.64	0.18 ↑	2.04
PC (16:0/18:1(9Z))	1.39 ↑	1.67	-	-	0.71 ↑	1.70	2.39 ↑	1.38
PC (15:0/22:4(7Z,10Z,13Z,16Z))	-	-	−1.64 ↓	1.89	−0.89 ↓	1.47	−0.88 ↓	1.55
LysoPC (20:4(5Z,8Z,11Z,14Z))	0.75 ↑	1.42	−1.20 ↓	1.84	-	-	-	-
LysoPC (20:1(11Z))	−3.15 ↓	1.96	−0.46 ↓	1.70	−0.34 ↓	1.63	−0.68 ↓	1.99
LysoPC (22:1(13Z))	−0.42 ↓	1.92	−0.30 ↓	1.80	-	-	-	-
LysoPC (14:0)	−0.78 ↓	1.57	−0.27 ↓	1.47	-	-	−0.42 ↓	2.02
LysoPC (18:1(9Z))	-	-	-	-	−0.23 ↓	1.70	−0.41 ↓	1.83
LysoPC (P-18:0)	-	-	-	-	-	-	0.27 ↑	1.64
Phosphocholine	0.53 ↑	1.68	-	-	-	-	-	-
Creatine	1.31 ↑	1.96	-	-	-	-	-	-
Leucine	−0.59 ↓	1.65	-	-	−0.47 ↓	2.27	−0.57 ↓	2.28
Anserine	−0.66 ↓	1.48	-	-	-	-	-	-
L-Valine	−1.54 ↓	1.53	-	-	-	-	-	-
Kynurenine	−0.54 ↓	1.54	0.64 ↑	1.61	-	-	-	-
L-Methionine	−0.62 ↓	1.79	-	-	0.37 ↑	1.52	-	-
3-Sulfino-L-alanine	-	-	−0.70 ↓	1.93	−0.66 ↓	2.06	-	-
Norvaline	-	-	-	-	−0.53 ↓	1.81	−0.76 ↓	2.13
L-Lysine	-	-	0.48 ↑	2.08	0.33 ↑	1.50	0.36 ↑	1.63
Prolylhydroxyproline	−0.48 ↓	1.72	0.32 ↑	1.71	-	-	-	-
Taurocholic acid	−3.52 ↓	1.64	-	-	-	-	-	-
Tauroursodeoxycholic acid	−1.01 ↓	1.52	-	-	-	-	-	-
Glycocholic acid	−0.70 ↓	1.41	-	-	-	-	-	-
Pregnenolone	−1.74 ↓	1.51	-	-	-	-	-	-
Epiandrosterone	−0.73 ↓	1.69	0.66 ↑	1.43	-	-	-	-
Androsterone	−0.64 ↓	1.49	0.76 ↑	1.58	-	-	-	-
Cortolone-3-glucuronide	−0.88 ↓	1.74	-	-	-	-	-	-
Cholesteryl acetate	−0.99 ↓	1.40	-	-	-	-	-	-
Dehydroepiandrosterone sulfate	-	-	0.64 ↑	1.83	-	-	-	-
4,4-Dimethylcholesta-8,14,24-trienol	-	-	1.16 ↑	1.61	-	-	-	-
Sphingosine	−0.46 ↓	1.25	0.80 ↑	1.57	-	-	0.69 ↑	1.96
Sphinganine 1-phosphate	−0.55 ↓	1.43	0.54 ↑	1.70	0.51 ↑	1.46	-	-
Phytosphingosine	-	-	-	-	−0.52 ↓	1.86	-	-
CerP (d18:1/22:0)	-	-	-	-	0.75 ↑	2.27	0.75 ↑	2.19
Cer (d18:0/12:0)	-	-	0.61 ↑	1.45	0.68 ↑	1.87	0.84 ↑	2.05
Cer (d18:0/16:0)	-	-	-	-	0.54 ↑	1.73	0.71 ↑	1.87
Retinoic acid	0.80 ↑	1.83	0.70 ↑	1.74	-	-	0.37 ↑	1.60
Biotin	-	-	−0.49 ↓	1.53	−0.37 ↓	1.59	-	-
MG (0:0/18:3(6Z,9Z,12Z)/0:0)	-	-	1.14 ↑	1.60	1.22 ↑	2.03	-	-
Choline	−0.30 ↓	0.65	-	-	−0.57 ↓	1.52	−1.09 ↓	2.11
Theophylline	-	-	−0.42 ↓	1.70	-	-	-	-
N-Acetyl-6-O-L-fucosyl-D-glucosamine	-	-	−1.78 ↓	2.31	−1.70 ↓	2.30	−1.46 ↓	1.87
Uracil	-	-	-	-	−0.20 ↓	1.47	−0.52 ↓	2.14

Note: ^a^ Log2 fold change (Log2FC) was obtained by the Wilcoxon test. ^b^ variable importance in the projection (VIP) was obtained from OPLS-DA analysis; the threshold value was 1.0. ↑ means up-regulation; ↓ means down-regulation. *n* = 6 in each group. LFD: low-fat-diet group; HFD: high-fat-diet group; EA: ellagic acid intervention group; EABC77: EA + *Weizmannia coagulans* BC77 intervention group; EABC2000: EA + *Weizmannia coagulans* BC2000 intervention group.

## Data Availability

The raw data of 16S rRNA gene libraries generated during this study are publicly available at the Sequence Read Archive (SRA) portal of NCBI under accession number PRJNA856975.

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
