# Peer review of "Supplementation of Weizmannia coagulans BC2000 and Ellagic Acid Inhibits High-Fat-Induced Hypercholesterolemia by Promoting Liver Primary Bile Acid Biosynthesis and Intestinal Cholesterol Excretion in Mice"

_microorganisms, 2023, doi:10.3390/microorganisms11020264_

Round 1

Reviewer 1 Report

xThis rodent-model study analyzed the inhibitory results and the mechanisms of the combination of probiotic Weizmannia coagulans BC2000 and ellagic acid on hyperlipidemia-induced cholesterol metabolism disorders, and examined the effects of BC2000 on the EA gut-metabolites. It is a continuation of a previous study which revealed that W. coagulans BC2000 increased the abundance of intestinal EA-transforming bacteria and inhibited hyperlipidemia-caused metabolic disorders.

The study addressed an important topic and it was well conducted. I would recommend this manuscript for publication after the following suggestions have been attended to:

Abstract: an introductory word should be used, such as “the probiotic Weizmannia coagulans (W. coagulans)

W. coagulans” should be used in the whole manuscript

Line 24: “iso-urolithin A”

Line 91: “.../mouse”

Discussion (Lines 393-410): the following ideas can be added here:

Gordonibacter urolithinfaciens converted ellagic acid into urolithins M5, M6, and C, and was positively correlated with the metabotype A. The recently discovered Ellagibacter isourolithinifaciens was also able to produce iso-urolithin A and positively correlated with metabotype B (García-Villalba et al. Metabolism of different dietary phenolic compounds by the urolithin-producing human-gut bacteria Gordonibacter urolithinfaciens and Ellagibacter isourolithinifaciens. Food Funct. 2020;11(8):7012-7022. doi: 10.1039/d0fo01649g).

- metabotype A includes producers of only urolithin A, metabotype B includes subjects who produce urolithin A, urolithin B, and iso-urolithin A, and metabotype 0 includes individuals that cannot produce final urolithinsEllagic acid and urolithins showed complex biological actions including the modulation of many important signaling pathways involved in inflammation and aging. Thus, metabolic phenotype 0 could mostly benefit from diet-addition of ellagic acid and probiotic Gordonibacter urolithinfaciens (Banc et al. The Impact of Ellagitannins and Their Metabolites through Gut Microbiome on the Gut Health and Brain Wellness within the Gut-Brain Axis. Foods 2023, 12, 270; doi: 10.3390/foods12020270).

- another study confirmed that gut bacterium Gordonibacter urolithinfaciens metabolized ellagic acid into urolithins M5, M6, and C through dehydroxylation with the involvement of NADPH and FAD (Watanabe et al. Evaluation of electron-transferring cofactor mediating enzyme systems involved in urolithin dehydroxylation in Gordonibacter urolithinfaciens DSM 27213. J Biosci Bioeng. 2020;129(5):552-557. doi: 10.1016/j.jbiosc.2019.11.014).

Lines 398-399: “Eggerthellaceae” should be deleted before Gordonibacter urolithinfaciens as Eggerthellaceae is the family

Line 404: please be specific as several species belong to Eggerthellaceae family

Line 405: the same

Line 409: the same

Author Response

Dear Reviewer,

Thank you very much for your valuable comments. We have made changes carefully to the text according to the comments and suggestions.

Here are the details of all the changes we have made to the manuscript in response to reviewers’ comments. All the changes are highlighted in yellow in the revised manuscript.

Reviewer 1

This rodent-model study analyzed the inhibitory results and the mechanisms of the combination of probiotic Weizmannia coagulans BC2000 and ellagic acid on hyperlipidemia-induced cholesterol metabolism disorders, and examined the effects of BC2000 on the EA gut-metabolites. It is a continuation of a previous study which revealed that W. coagulans BC2000 increased the abundance of intestinal EA-transforming bacteria and inhibited hyperlipidemia-caused metabolic disorders.

The study addressed an important topic and it was well conducted. I would recommend this manuscript for publication after the following suggestions have been attended to:

1: Abstract: an introductory word should be used, such as “the probiotic Weizmannia coagulans (W. coagulans)”.

Re: Revised (L15).

2.“W. coagulans” should be used in the whole manuscript

Re: Checked the whole manuscript. "W. coagulans" was used in the whole manuscript.

  1. Line 24: “iso-urolithin A”

Re: Revised (L25).

  1. Line 91: “.../mouse”

Re: Revised (L92-94).

  1. Discussion (Lines 393-410): the following ideas can be added here:

Gordonibacter urolithinfaciens converted ellagic acid into urolithins M5, M6, and C, and was positively correlated with the metabotype A. The recently discovered Ellagibacter isourolithinifaciens was also able to produce iso-urolithin A and positively correlated with metabotype B (García-Villalba et al. Metabolism of different dietary phenolic compounds by the urolithin-producing human-gut bacteria Gordonibacter urolithinfaciens and Ellagibacter isourolithinifaciens. Food Funct. 2020;11(8):7012-7022. doi: 10.1039/d0fo01649g).

- metabotype A includes producers of only urolithin A, metabotype B includes subjects who produce urolithin A, urolithin B, and iso-urolithin A, and metabotype 0 includes individuals that cannot produce final urolithins. Ellagic acid and urolithins showed complex biological actions including the modulation of many important signaling pathways involved in inflammation and aging. Thus, metabolic phenotype 0 could mostly benefit from the diet-addition of ellagic acid and probiotic Gordonibacter urolithinfaciens (Banc et al. The Impact of Ellagitannins and Their Metabolites through Gut Microbiome on the Gut Health and Brain Wellness within the Gut-Brain Axis. Foods 202312, 270; doi: 10.3390/foods12020270).

- another study confirmed that gut bacterium Gordonibacter urolithinfaciens metabolized ellagic acid into urolithins M5, M6, and C through dehydroxylation with the involvement of NADPH and FAD (Watanabe et al. Evaluation of electron-transferring cofactor mediating enzyme systems involved in urolithin dehydroxylation in Gordonibacter urolithinfaciens DSM 27213. J Biosci Bioeng. 2020;129(5):552-557. doi: 10.1016/j.jbiosc.2019.11.014).

Re: Revised (L416-L427).

  1. Lines 398-399: “Eggerthellaceae” should be deleted before Gordonibacter urolithinfaciensas Eggerthellaceae is the family

Re: Revised (L420-L426).

  1. Line 404: Please be specific as several species belong to Eggerthellaceae family

Re: Revised (L427-429).

  1. Line 405: the same

Re: Revised (L427-L429).

  1. Line 409: the same

Re: Revised (L431-L432).

Reviewer 2 Report

- Methods are detailed and well-explained.

-Titles of each subsection in the Results section should be substitute by a sentence stating the results obtained.

- A concluding sentence is needed at the end of each subsection in the Results section.

- The Introduction and the Discussion provide sufficient information to understand the state-of-the-art and citations are appropiate.

Author Response

Dear Reviewer,

Thank you very much for your valuable comments. We have made changes carefully to the text according to the comments and suggestions.

Here are the details of all the changes we have made to the manuscript in response to reviewers’ comments. All the changes are highlighted in yellow in the revised manuscript.

Reviewer 2

Comments and Suggestions for Authors

  1. Methods are detailed and well-explained.

Re: Thank you for your comments on this article.

  1. Titles of each subsection in the Results section should be substitute by a sentence stating the results obtained.

Re: Revised (L195, L226, L244, L277, L294,L325, L362).

  1. A concluding sentence is needed at the end of each subsection in the Results section.

Re: Revised (L210-215, L235-237, L264-265, L285-286, L309-311,L348-350, L370-L372).

  1. The Introduction and the Discussion provide sufficient information to understand the state-of-the-art and citations are appropiate.

Re: Thank you for your comments on this article.
